# Research on a Dynamic Algorithm for Cow Weighing Based on an SVM and Empirical Wavelet Transform

**DOI:** 10.3390/s20185363

**Published:** 2020-09-18

**Authors:** Ningning Feng, Xi Kang, Haoyuan Han, Gang Liu, Yan’e Zhang, Shuli Mei

**Affiliations:** 1Key Laboratory of Modern Precision Agriculture System Integration Research, Ministry of Education, China Agricultural University, Beijing 100083, China; s20183081297@cau.edu.cn (N.F.); B20183080643@cau.edu.cn (X.K.); pac@cau.edu.cn (G.L.); 2Key Laboratory of Agricultural Information Acquisition Technology, Ministry of Agriculture, and Rural Affairs, China Agricultural University, Beijing 100083, China; 3College of Information and Electrical Engineering, China Agricultural University, Beijing 100083, China; sy20193081512@cau.edu.cn (H.H.); meishuli@cau.edu.cn (S.M.)

**Keywords:** cow, dynamic weighing, SVM, motion state, empirical wavelet transform

## Abstract

Weight is an important indicator of the growth and development of dairy cows. The traditional static weighing methods require considerable human and financial resources, and the existing dynamic weighing algorithms do not consider the influence of the cow motion state on the weight curve. In this paper, a dynamic weighing algorithm for cows based on a support vector machine (SVM) and empirical wavelet transform (EWT) is proposed for classification and analysis. First, the dynamic weight curve is obtained by using a weighing device placed along a cow travel corridor. Next, the data are preprocessed through valid signal acquisition, feature extraction, and normalization, and the results are divided into three active degrees during motion for low, medium, and high grade using the SVM algorithm. Finally, a mean filtering algorithm, the EWT algorithm, and a combined periodic continuation-EWT algorithm are used to obtain the dynamic weight values. Weight data were collected for 910 cows, and the experimental results displayed a classification accuracy of 98.6928%. The three algorithms were used to calculate the dynamic weight values for comparison with real values, and the average error rates were 0.1838%, 0.6724%, and 0.9462%. This method can be widely used at farms and expand the current knowledgebase regarding the dynamic weighing of cows.

## 1. Introduction

The weight of cows is an important parameter for their growth and development [1,2,3]. Notably, weight provides an important basis for breeding staff to determine the production performance of cows, for implementing appropriate feeding strategies [4,5,6], for determining nutrition [7], and for establishing medicine doses. Weight is also an important reference index related to the age of mating, reproductive performance [8,9,10], and the time of delivery [11,12]. Weight variations can also provide early warning for some diseases [13]. Therefore, conveniently and accurately obtaining the weight of cows is important in animal husbandry.

The traditional weighing method is static weighing, and it involves driving a cow onto a measuring device with a fence around it. The cow must remain still for a period of time [14] before a stable weight value can be read. Generally, the weight of cows can be accurately obtained. However, large individual cows usually need to be driven by more than one person, which is dangerous to some extent [15]. Moreover, such an operation will cause stress reactions in cows, resulting in negative effects, such as a decreased milk yield. To avoid man-made interference, Pastell et al. [16] used a milking robot equipped with weighing equipment for cow weight in a stationary state, and this approach had a good effect on standing normal cows. However, if cows are leaning against or not standing completely on the platform, there will be a large weighing error, and some farms do not have the appropriate installation conditions for such structures. Therefore, some scholars studied dynamic weighing modes [17]. Weighing equipment is installed along the only path that animals can use, such as the entrance to a cowshed or milking hall. Dickinson et al. [18] built a gait-based weighing system that combined an RFID (radio frequency identification system) and automatic weighing equipment. They compared the dynamic weights with actual static weights, which confirmed the feasibility and effectiveness of the dynamic weighing method in the evaluation of cow weights. Alawneh et al. [19] recorded the daily live weight variations in 463 cows at a dairy farm in northern New Zealand, and the standard deviation of the measured value was 17 kg; they obtained the live weights of cows through autocorrelation analysis, and these weights were used to adjust the cattle feeding plan and detect changes in the bodies of cows. E. Gonzalez-Garcia et al. [20] designed a mobile automatic weighing system based on an environment with outdoor free-range sheep and encouraged the voluntary weighing of sheep combined with stimulations of water and mineral salt intake; this approach not only saved human and material resources but also improved animal productivity and welfare. The above research showed that a dynamic weighing device does not require human intervention and can automatically obtain weight data when placed along a given pathway. However, an automatic weighing device obtains a dynamic weight curve when cows are walking, and there are generally errors associated with weight values obtained from the curve. To reduce such errors as much as possible, some improved methods have been reported. The initial processing algorithm is the simplest mean filtering method, but when cows move quickly, the error of this algorithm will be large. Cveticanin et al. [21] simplified the four-legged movement system of cows into a bipedal system similar to that of humans and built a model to measure the weights of cows on the basis of their walking speed and the weights of objects; as long as animal crowding on the platform is avoided, the error rate can reach less than 2%. Wen et al. [22] established a Back Propagation (BP) neural network model by using the output signal of the weight sensor and the animal movement speed to obtain the animal weight; this method laid the foundation for the application of BP neural networks in dynamic weighing algorithms. Larios et al. [23] studied a small kestrel breeding colony in southern Spain and used an artificial neural network method to process a large number of dynamic weight data for birds to estimate the weights of individual birds. The results showed that the system has a low cost and does not require human observers; thus, this approach is friendly to wild birds. Wang et al. [24] proposed a dynamic weighting method based on wavelet denoising and empirical mode decomposition (EMD). The sampled signal was subjected to wavelet denoising before performing EMD, and the residual normalized result can be used to obtain the weight value. The EMD algorithm, which displayed excellent adaptability in dynamic weighing, obtained very good results but lacked theoretical support; additionally, endpoint effects were observed [25]. Therefore, this approach must be improved.

When a large cow walks on a weighing platform, any slight movement will have an impact on the waveform of the weight curve. The waveforms of different active degrees during traveling can considerably vary. Therefore, a single algorithm is generally adopted to analyze the weight curves of animals in different moving states, and this approach has certain limitations. At present, there are many dynamic weighing algorithms, but most of them do not consider the active degree of the animal being weighed.

In this paper, for the first time, a method based on the SVM classification algorithm and EWT for the automatic dynamic weighing of cows is presented. We collected the dynamic weight curve of cows by placing the weighing equipment along a path used by cows. The weight curve was used to obtain five eigenvalues that express signal information. The eigenvalues were classified by SVM to obtain three active degrees of cows: low grade, medium grade, and high grade. According to the different exercise statuses of the cows, different methods were selected for weight measurement, and finally, the dynamically determined weight of each cow was obtained.

## 2. Materials and Methods

### 2.1. Dynamic Signal Acquisition

The experiment took place on a farm in northern China with more than 500 Holstein cows. The animals go to a milking hall that can hold 32 cows at a time each day, and our laboratory equipment was located in the exit passageway of the milking hall. The experiment lasted for a week and was performed at the same time every day. The field situation is shown in Figure 1. All experimental operations were allowed. Because the familiarity of the cows with the environment suddenly changed, a few days before the experiment officially started, we installed equipment during nonmilking intervals to give the cows time to get used to the weighing environment.

During the experiment, guardrails were placed at each end of the device to ensure that only one cow passed at a time. Two weighing body were placed side by side and calibrated using the same method. There are four weighing sensors in the scale body, which are installed at four support points on the weighing table. The type of the sensor is SQB-500, and its rated load is 500 kg. The accuracy class is C4, and the sensitivity is 3.0 ± 0.003 mv/V. One platform was used to obtain a dynamic weight signal, and the other was used to obtain the true weight. Notably, obstacles were established at the end of the second weighing platform to make cows stay on the weighing platform long enough to obtain their weight at that time. Several consecutively obtained values were used to calculate the average, which was deemed the real cow weight. A schematic diagram of the weighing experiment is shown in Figure 2. Each scale has a length of 250 cm, a width of 80 cm, and a height of 10 cm. The distance of 80 cm ensures that the cows will not touch the guardrail or become crowded. The weighing equipment was placed before and after the sloped areas to facilitate cow movement up and down. A gap of 1–2 cm was left between each weighing instrument and the slope to avoid the error caused by the collision of parts due to the shaking of the weighing table during cow movement. There were video recordings of cow progress, and such a video can aid in the subsequent calibration of the motion state. The weighing device communicated with the upper computer via an RS485 link, and the sampling interval was set to 100 ms. During the experiment, the software was refreshed every time a cow passed on the device to ensure that the net weight was always 0 when no cows were on the scale, thus eliminating the interference of cow dung and urine on the weighing platform. A total of 910 weighing curves were collected in this experiment, and the processing method of the weighing data will be given in the next section.

### 2.2. Dynamic Weighing Algorithm

#### 2.2.1. Preprocessing of the Dynamic Weight Signal

The collected weight information was plotted as the original weight curve W, and the results are shown in Figure 3a; notably, the dynamic weighing of cows involves three stages: The first half of a cow is on the weighing platform, the entire cow is on the weighing platform, and the second half of a cow is on the weighing platform (i.e., after the front half steps off the scale). All four hooves were required to be on the weighing platform to obtain valid data. The effective data acquisition process was as follows:
Set the threshold $K_{1}=\eta \cdot W_{max}$, and set η=0.8 [26]. When a weight is less than $K_{1}$, the value is set to 0, and the resulting graph is shown in Figure 3b.Take the derivative of the function in Figure 3b and obtain Figure 3c.Find the starting positions of the rising edge and the falling edge, i.e., points A and B; the data between A and B are valid data, as shown in Figure 3d.


The effective signal has different lengths, so it is inconvenient to use it as the *x* input of the classifier. To unify the standard, the characteristic information contained in the signal needs to be obtained. In this paper, five characteristics were used to express signal information, namely, the range R, standard deviation σ, peak factor PAR, waveform factor FF, and velocity v. The formula for each eigenvalue is as follows:
(1)$$R=f_{max}-f_{min},$$
where $f_{max}$ is the maximum value in the valid data set and $f_{min}$ is the minimum value in the valid data set:
(2)$$\sigma =\sqrt{\frac{\sum_{i=1}^{N}{\left(f\left(i\right)-\bar{f}\right)}^{2}}{N}},$$
where f(i) represents the valid data, $\bar{f}$ is the average of the valid data, and N is the number of valid values:
(3)$$PAR=\frac{R}{\sqrt{\frac{\sum_{i=1}^{N}{\left(f\left(i\right)\right)}^{2}}{N}}}.$$


In the formula, R is the peak (range):
(4)$$FF=\frac{\sqrt{\frac{\sum_{i=1}^{N}{\left(f\left(i\right)\right)}^{2}}{N}}}{\bar{f}},$$
(5)$$v=\frac{2.5}{t},$$
where t is the time required for cows to walk across a weighing platform with a length of 2.5 m, as shown in Figure 3a.

In statistical analysis, some eigenvalues much differ in an order of magnitude, and those with large values will dominate in the process of model construction and lead to slow convergence in the optimization process, which is not conducive to performing the subsequent calculations. To improve the accuracy and convergence speed of the model, it is necessary to normalize the eigenvalues. The corresponding formula is as follows:
(6)$$X=\frac{x-x_{min}}{x_{max}-x_{min}},$$
where x is the eigenvalue data, $x_{min}$ is the minimum eigenvalue, and $x_{max}$ is the maximum eigenvalue.

#### 2.2.2. SVM Classification

The motion state of cows directly affects the waveform changes in the effective signals. When the cow has a high active degree during moving, the waveform amplitude is large, the overall weighing performance is variable, and the weighing duration is short. When the active degree is low, the waveform amplitude is small, the overall weighing performance is relatively stable, and the weighing duration is comparatively long. Different active degrees lead to different waveforms, and using the same method to analyze the waveforms in different active degrees is too singular and limited, thereby increasing the error rate of measurements. The two-stage method of classification and analysis can be adopted to improve the robustness of the algorithm.

In the experiment, dynamic weight data were collected for 910 cows. According to the comprehensive consideration of five characteristics, the active degree of cows was divided into three grades: low, medium, and high. The videos of the four hooves of cows walking on the weighing platform were used to obtain the effective frame counts, and then the effective weighing time $t_{v}$ was calculated according to the frame counts. Two experts were invited to evaluate and score the state of the cows with information, such as the mental state of the cows, the health of the hooves, and the height of the leg lift. The scores were recorded as grade 0, grade 1, and grade 2. We collected all the effective weighing time $t_{v}$, and set the threshold value $t_{1}$, $t_{2}$ (the threshold value was not fixed, and fluctuated according to the expert score). Considering $t_{v}$ and the scores, each collected data subset was demarcated as 0, 1, or 2 corresponding to a low, medium, or high grade of active degree, respectively. The characteristic average values of the three states were calculated as shown in Table 1. Notably, the eigenvalue distribution is the smallest in the slow state, the largest in the fast state, and moderate in the medium state. The three states have certain differences based on different eigenvalues. After combining the five features, the SVM algorithm was used to classify the data. The specific classification process will be described in detail in the following section.

First, dataset $D=\left\{\left(x_{1},y_{1}\right),\left(x_{2},y_{2}\right),\ldots ,\left(x_{m},y_{m}\right)\right\}$ was constructed, where $x_{i}=\left(R_{i},\sigma_{i},PAR_{i},FF_{i},v_{i}\right)$ is a feature vector and $y_{i}\epsilon \left\{0,1,2\right\}$ represents the tags for the active degrees. There is commonly a problem with class imbalance in data sets. SMOTE is used to extend the minority sample. Then, the data set is randomly [27] divided into a training set and a test set. The SVM algorithm can appropriately divide the data set based on its inherent characteristics, and the anti-noise performance is represented as a highly robust hyperplane [28] $f\left(x\right)=\omega^{T}\phi \left(x\right)+b$. In this case, ω is the normal vector, b is displacement, and ϕ(x) is mapped *x* to the high-dimensional feature space of the feature vector. For the training set $\left\{\left(x_{1},y_{1}\right),\left(x_{2},y_{2}\right),\ldots \ldots ,\left(x_{m},y_{m}\right)\right\}$, where $y_{i}\epsilon \left\{0,1,2\right\}$, the SVM optimization objective is:
$$\underset{\omega ,b,\xi_{i}}{\min}\frac{1}{2}{\left\|\omega \right\|}^{2}+C\sum_{i=1}^{m}\xi_{i},$$
(7)$$s.t.\;y_{i}\left(\omega^{T}\phi \left(x_{i}\right)+b\right)\geq 1-\xi_{i},i=1,2,\cdots ,m.$$


In this case, C>0 is the penalty factor of the error term, and $\xi_{i}\geq 0$ is the relaxation variable.

To address optimization objective (7), the following classification function can be obtained:
(8)$$L\left(x\right)=sign\left(\sum_{i=1}^{n}\alpha_{i}y_{i}\kappa \left(x_{i},x\right)+b\right),$$
where $\alpha_{i}$ is the Lagrangian factor and $\kappa \left(x_{i},x\right)$ is a kernel function that maps the feature vector to the high-dimensional space. According to the statistical findings, the five features conform to the Gaussian distribution, so the RBF was selected as the kernel function; this function can be formulated as follows:
(9)$$\kappa \left(x_{i},x_{j}\right)=\exp\left(-\frac{{\left\|x_{i}-x_{j}\right\|}^{2}}{2\sigma^{2}}\right),$$
where ($\kappa \left(x_{i},x_{j}\right)=<\phi \left(x_{i}\right),\phi \left(x_{j}\right)>=\phi {\left(x_{i}\right)}^{T}\phi \left(x_{j}\right)$, σ>0 is the bandwidth of the Gaussian kernel.

Here:
(10)$$g=\frac{1}{2\sigma^{2}}.$$


The penalty factor C and kernel function parameter G are the keys to the classification of the dynamic weight signals for cows. These values influence the quality of the SVM model. In this paper, grid search and cross-validation [29] methods were used to find the optimal C and G values.

#### 2.2.3. Dynamic Weight Value Calculation

Above, the data were divided into three categories according to the active degree, and three algorithms were adopted to process the data and finally calculate the dynamic body weight $m_{c}$.

1. Weight value calculation at low-grade activity

Figure 4a,b show the original weight signal and the extracted valid signal at a low-grade activity. After analysis, it was found that the amplitude fluctuation range of the effective signal in the low grade is small, the waveform duration is the longest among those for the three states, the overall performance is stable, and the waveform does not considerably deviate from that in the normal state.

The improved mean filtering algorithm was used to process the signal in the low grade of active degree. The algorithm procedure was as follows: The maximum and minimum values in the effective data sequence f(t) are removed, and the sequence $f{\left(t\right)}^{\prime }$ is obtained. The average value of the sequence is used to obtain the weight value $m_{c}$ under dynamic weighing conditions, as shown in the following formula:
(11)$$m_{c}=\frac{\sum_{i=1}^{n1}f{\left(t\right)}^{\prime }}{n1},$$
where n1 is the number of data points in the signal sequence $f{\left(t\right)}^{\prime }$.

2. Weight value calculation at medium-grade activity

The amplitude fluctuations in the medium grade of active degree dynamic weight signal are larger than those for the low-grade signal, the waveform variations are more obvious, and the waveform duration is slightly shorter. The original weight signal and effective signal in this state are shown in Figure 5a,b. According to the nonstationary and nonlinear characteristics of such signals, the EWT algorithm can be used to process the effective data. The EWT algorithm [30] is adaptive [31], and a tightly supported AM-FM signal component can be extracted according to the characteristics of the effective signal of the cow’s dynamic weigh [32]. A single effective signal is decomposed into multiple modal components with different frequency characteristics; one of the components is associated with the residual volume, which can reflect the trend of the effective signal or mean and be used to obtain the dynamic weight value $m_{c}$. The EWT decomposition of the effective weight signal f(t) of cows at medium speed can be expressed as follows:
Calculate the Fourier spectrum F(w) of the effective signal f(t).Normalize the Fourier spectrum to the interval [0,π] and divide the spectrum into *N* intervals; in addition to the boundary value 0 and π, there are *N* − 1 boundaries that need to be determined. $w_{n}$ is defined as a boundary, and the minimax method for spectral searches is used to determine the other boundaries. Then, each segment can be expressed as:
(12)$$\Lambda_{n}=\left[w_{n-1},w_{n}\right]n=1,2,\ldots \ldots ,N,\mathrm{and}\cup_{n=1}^{N}\Lambda_{n}=\left[0,\pi \right].$$
The empirical wavelet is a band-pass filter defined on the interval $\Lambda_{n}$ based on the construction of Littlewood-Paley and Meyer theories, the wavelet scale function ${\hat{\phi }}_{m}\left(w\right)$ and wavelet function ${\hat{\psi }}_{n}\left(w\right)$ can be defined as follows:
(13)$${\hat{\phi }}_{m}\left(w\right)=\{\begin{array}{c} 1,\left|w\right|\leq \left(1-\gamma \right)w_{m}\\ \cos\left[\frac{\pi }{2}\beta \left(\frac{1}{2\gamma w_{m}}\left(\left|w\right|-\left(1-\gamma \right)w_{m}\right)\right)\right]\\ 0,\mathrm{other} \end{array},\left(1-\gamma \right)w_{m}\leq \left|w\right|\leq \left(1+\gamma \right)w_{m},$$
(14)$${\hat{\psi }}_{n}\left(w\right)=\{\begin{array}{c} 1,\left(1+\gamma \right)w_{m}\leq \left|w\right|\leq \left(1-\gamma \right)w_{m+1}\\ \cos\left[\frac{\pi }{2}\beta \left(\frac{1}{2\gamma w_{m+1}}\left(\left|w\right|-\left(1-\gamma \right)w_{m}\right)\right)\right],\left(1-\gamma \right)w_{m-1}\leq \left|w\right|\leq \left(1+\gamma \right)w_{m+1}\\ \sin\left[\frac{\pi }{2}\beta \left(\frac{1}{2\gamma w_{m}}\left(\left|w\right|-\left(1-\gamma \right)w_{m}\right)\right)\right],\left(1+\gamma \right)w_{m}\leq \left|w\right|\leq \left(1-\gamma \right)w_{m} \end{array},$$
where $\gamma <min_{m}\left(\frac{w_{m+1}-w_{m}}{w_{m+1}+w_{m}}\right)$ and $\beta \left(x\right)=x^{4}\left(35-84x+70x^{2}-20x^{3}\right).$Calculate EWT through the construction method of another wavelet transform. The specific formulas of the detailed function $w_{f}^{\varepsilon }\left(n,t\right)$ and the approximate function $w_{f}^{\varepsilon }\left(0,t\right)$ are as follows:
(15)$$w_{f}^{\varepsilon }\left(n,t\right)=<f\left(t\right),\psi_{n}\left(t\right)>=\int f\left(\tau \right)\bar{\psi_{n}\left(\tau -t\right)}d\tau =F^{-1}\left[f\left(w\right)\hat{\psi_{n}}\left(w\right)\right],$$
(16)$$w_{f}^{\varepsilon }\left(0,t\right)=<f\left(t\right),\varphi_{1}\left(t\right)>=\int f\left(\tau \right)\bar{\varphi_{1}\left(\tau -t\right)}d\tau =F^{-1}\left[f\left(w\right)\hat{\psi_{1}}\left(w\right)\right],$$
where $\psi_{n}\left(t\right)$ represents the EWT; $\hat{\psi_{n}}\left(w\right)$ and $\hat{\psi_{1}}\left(w\right)$ are the Fourier transform of $\psi_{n}\left(t\right)$ and $\psi_{1}\left(t\right)$; $\bar{\psi_{n}\left(t\right)}$ and $\bar{\varphi_{1}\left(t\right)}$ are the complex conjugate functions of $\psi_{n}\left(t\right)$ and $\varphi_{1}\left(t\right)$; and <,> denotes an inner product operation.After obtaining the wavelet detail coefficient and approximate coefficient through the first four steps, for the effective signal f(t), the mathematical expression after decomposition is as follows:
(17)$$f\left(t\right)=\sum_{i=0}^{N}f_{i}\left(t\right),$$
where $f_{i}\left(t\right)$ is the decomposed AM-FM component and *N* is the number of decomposed signal components. The expression of each component is shown in (18):
(18)$$\{\begin{array}{c} f_{k}\left(t\right)=w_{f}^{\varepsilon }\left(n,t\right)\psi_{k}\left(t\right)\\ f_{0}\left(t\right)=w_{f}^{\varepsilon }\left(0,t\right)\varphi_{1}\left(t\right) \end{array}.$$



The components of the effective dynamic weight signal obtained with the EWT algorithm are shown in Figure 6. $f_{k}\left(t\right)\left(k=1,2\right)$ reflects the high-frequency part of the effective weight signals for cows and encompasses the local characteristics of the effective signal. The signal component $f_{0}\left(t\right)$ reflects the low-frequency part of the effective signal and the trend or mean value of the signal. The weight value $m_{c}$ of the medium grade under dynamic weighing conditions can be obtained through the calculation of $f_{0}\left(t\right)$:
(19)$$m_{c}=\frac{\sum_{i=1}^{m}f_{0}\left(t\right)}{m},$$
where *m* is the number of signal components $f_{0}\left(t\right)$.

3. Weight value calculation at high-grade activity

The original weight signal and the extracted valid signal in the high grade of active degree are shown in Figure 7a,b. The amplitude fluctuation range of the valid signal is the largest among those at other grades, the waveform duration is short, the whole wave fluctuates, and the waveform largely deviates from the under a normal state. In this case, the cow walks onto the weighing platform for a short time, and the walking speed is relatively fast.

Compared with the signals in the low-grade and medium-grade active degrees, the signal length in the vigorous motion state is shorter, the signal fluctuation is larger, and the signal information cannot be well interpreted. Therefore, the effective signal sequence is regarded as a periodic signal for periodic continuation. Assuming that the length of the wavelet filter is *M*, the length of the effective signal sequence F (*t*) is *N*, and the extended signal is $f_{d}\left(t\right)$, then:
(20)$$f_{d}\left(t\right)=\{\begin{array}{cc} f\left(t\right), & 0\leq t\leq N-1\\ f\left(t-N\right), & N-1\leq t\leq N+M-2 \end{array}.$$


EWT decomposition is performed by using the extended signal $f_{d}\left(t\right)$. The decomposition process used is consistent with the decomposition process applied for the effective signal at the medium grade and will not be redescribed here. The decomposition result is shown in Figure 8.

The signal component $f_{0}\left(t\right)$ decomposed by the EWT algorithm reflects the trend or mean value of the effective signal for the dynamic weighing of cows in the high grade of active degree. The weight value $m_{c}$ can be obtained through calculating $f_{0}\left(t\right)$, and the corresponding formula is as follows:
(21)$$m_{c}=\frac{\sum_{i=1}^{m}f_{0}\left(t\right)}{m},$$
where *m* is the number of signal components $f_{0}\left(t\right)$.

#### 2.2.4. Evaluation

1. Classification evaluation index

In general, we used the accuracy rate to evaluate the classification model because it is a relatively intuitive evaluation index. The accuracy formula is as follows:
(22)$$\mathrm{Accuracy}=\frac{\mathrm{TP}+\mathrm{TN}}{\mathrm{TP}+\mathrm{TN}+\mathrm{FP}+\mathrm{FN}},$$
where TP, FN, FP, and TN are the components of the classification confusion matrix; specifically, TP represents the number of correct classifications for positive samples, FN represents the number of incorrect classifications for positive samples predicted to be negative samples, FP represents the number of incorrect classifications for negative samples predicted to be positive samples, and TN represents the number of correct classifications for negative samples [33].

However, using accuracy alone is not sufficient for fully evaluating the classification effect. To better evaluate the classification model, four evaluation indexes were considered: precision, recall, specificity, and the F1_score. The formulas for these indexes are shown in Table 2.

2. Weight as an evaluation index

By comparing the dynamic weight value with the real value, the accuracy of the weight algorithm can be assessed. Therefore, the error, which reflects the difference between the real and predicted values, was selected as the evaluation index. According to the dynamic weight value $m_{c}$ and real value $m_{s}$, the weight estimation error can be calculated as follows (body weight estimation errors):
(23)$$error=\frac{\left|m_{c}-m_{s}\right|}{m_{s}}\times 100\%.$$


Taking the weight data of the cow whose ID number is 130314 as an example, the acquisition process of real values $m_{s}$ is illustrated. With the obstacle placed at the end of the second weighing platform, the cow will be stagnant for a certain period of time. The weighing instrument can read that weight at this moment, as shown in the orange box in Figure 9. The average value of the collected data is calculated as $m_{s}$ and treated as the true value of the cow weight. The $m_{s}$ value of this cow is 602.2 kg.

The dynamic weighing curve corresponding to this cow is shown in the orange curve in Figure 9. The SVM classifier divides its valid signal into the medium-grade activity, so the EWT algorithm is used to calculate its dynamic weighing value. Since the calculation process iis described in Section 2.2.3, it will not be repeated here. In Figure 9, the effective part of the original weighing curve is decomposed by the EWT algorithm to obtain the signal component $f_{0}\left(t\right)$. The dynamic weighing value $m_{c}$ can be obtained by calculating the average value of $f_{0}\left(t\right)$, which is 599.1 kg. According to formula (19), the dynamic weighing error rate of the cow with ID number 130314 is about 0.5148%.

## 3. Results and Discussion

### 3.1. SVM Classification Results

The classification of weight data with the SVM algorithm is a prerequisite for the accurate acquisition of the dynamic weight values of cows. In this section, the classification results are analyzed and discussed. The original data set for this experiment included 910 values; the unbalanced data were processed with the SMOTE algorithm [34], and the overall number of values increased to 1527. The data were randomly assigned to the training set and the test set at a ratio of 8:2 [35]. The training set was used to train the SVM model, and the test set was used for verification. According to the SVM classification results, the actual classification and predictive classification diagrams of the test set were obtained, as shown in Figure 10.

In Figure 10, a blue ○ represents a correct prediction based on the actual classification for the test set, a lavender * represents a correct prediction, a magenta □ represents an incorrect prediction based on the actual classification, and a red ▽ represents an incorrect prediction. Notably, four red labels and four magenta labels can be observed; the red triangles and magenta squares have a longitudinal one-to-one correspondence, which also means that there are two actual high grades mistaken for medium grades and two medium grades are mistakenly classified as low grades. In addition, other actual and predicted classifications are consistent, and most of the predicted and actual values are similar. According to the statistics for the actual and predicted values of each grade in the figure, the classification confusion matrix is obtained as shown in Table 3:

For the three different active degrees, Table 3 shows that the values of TP, FN, FP, and TN in the low grade are 101, 0, 2, and 203, respectively. At a medium grade, the values of TP, FN, FP, and TN are 88, 2, 2, and 214, respectively. In the high grade, the values of TP, FN, FP, and TN are 113, 2, 0, and 201, respectively. According to the distribution of the confusion matrix under the three grades, the overall accuracy, precision, recall, specificity, and F1_score values were calculated, as shown in Table 4.

Table 4 shows that the overall classification accuracy is 98.6928%, which reflects a good classification effect. Among the evaluation indexes for the three active degrees, the accuracy, specificity, and F1_score of the high grade of active degree are the highest, reaching 100%, 100%, and 0.9912, respectively. The accuracy, recall, and F1 score of the medium-grade state are the lowest at 97.7778%, 97.7778%, and 0.9778, respectively. The highest recall rate was observed for the low grade at 100%, but the specificity was the lowest at 99.0244%. Overall, the classification model designed in this paper has a good effect and can accurately distinguish among the three active degrees. The classification results for the medium state slightly differ from those for the low and high grades. The statistical results in this paper support the effectiveness of the proposed method, and the classification results are reasonable. A method for classifying the active degree before calculating the weight value has not been previously reported, so the proposed approach provides a reference for assessing the active degree during the traveling of cows in actual production.

### 3.2. Dynamic Weighing Results

After classification, the dynamically obtained weight data were divided into three classes: low grade, medium grade, and high grade of active degrees during walking. The mean error rate and maximum error rate were calculated for the three active degrees, and the results are shown in Table 5.

Table 5 shows the average error rate and maximum error rate for the low, medium, and high active degrees. The average error rate and maximum error rate are the largest at 0.9462% and 1.9696%, respectively, for the high-grade activity. For the low grade, the average error rate and maximum error rate are 0.1838% and 0.5742%, respectively, which are the lowest values obtained. The values for the medium grade are 0.6724% and 1.6924%, respectively. In general, the average error rate calculated by the dynamic weighing algorithm is less than 1%. The calculated values in the three active degrees are consistent with the actual values. In the high grade of active degree, cows move quickly, with severe waveform oscillation and less valid data, so the average error rate of the weight values is largest in this state.

Although the average error rate reflects a good calculation effect, there are still some cases with large error rates, and the maximum error rate in the high grade of active degree approaches 2% in some cases. Based on the dynamic weight data collected and the videos taken during the experiment, the reasons for the high error rate are as follows. During the walking process, cows bounce, sneeze, etc., leading to fluctuations in the data and subsequent effects on the calculated weights. In some cases, cows excrete feces or urine on the first weighing platform and then go to the second weighing platform. This situation can lead to a large difference between the two measured weights, which is also an important factor related to the large error rate. In addition, in some stimulations outside the system, the movement speed of cows will be faster than normal states, leading to the acquisition of less valid data and resulting in a large error rate.

To verify the reliability of the algorithm developed in this paper, the EMD algorithm, a commonly used dynamic weighing signal analysis method, was applied to the collected weight signals of cows, and the average error rate was calculated for the medium and high grades. The results of the EMD algorithm and the algorithm proposed in this study are compared in Figure 11.

According to the comparison and analysis in Figure 11, the average error rate calculated by the EWT algorithm is lower than the average error rate obtained by the EMD algorithm in the dynamic weighing processing of cows at medium and high grades. The error rate of the EWT algorithm is less than 1%, and the average error rate of the EMD algorithm is less than 1.15%. The EWT algorithm performs better than the EMD algorithm in processing the dynamically obtained weights of cows. In fact, the EWT algorithm is an improvement of the EMD algorithm that inherits the advantages of rigorous wavelet transform theory and solves the traditional mode alias and large computational load problems of the EMD algorithm [25]. Taking the dynamic weighing data of the dairy cow with ID number 130314 as an example, the running time of the EWT algorithm is 0.0589 s when processing data, but the EMD algorithm’s is 0.5458 s, a difference of nearly 10 times of order of magnitude. The dynamic weighing equipment of dairy cows is used in the breeding farm and the cow weight value in real time is needed, so the EWT algorithm with a shorter running time is more in line with actual production needs. Therefore, the EWT algorithm can be effectively applied to weight curve processing in medium and high active degrees.

## 4. Conclusions

In this study, we collected the dynamic weight curve of cows by placing the weighing equipment along a path used by cows and the data were preprocessed. An SVM classification model was used to classify the weight signals, and a classification method for three active degrees during the traveling of cows was proposed for the first time. According to the active degree, the improved mean filtering, EWT algorithms, and the EWT based on the periodic extension algorithm were used to obtain the dynamic weight values. The results showed that the accuracy of the established classification model is 98.6928%. The data can be effectively divided into three categories according to the cow active degrees to provide a good basis for the accurate calculation of body weight. Three different algorithms were used to obtain dynamic weight values, and the average error rates were 0.1838%, 0.6724%, and 0.9462%, respectively, with an average error rate of less than 1%; thus, the proposed method meets the actual management demands. The research in this paper has practical value, and the developed method can be widely applied at farms to greatly reduce the manpower, material resource, and financial resource requirements. However, the error rate may be large in some cases, and research will be continued in the future to improve the accuracy of the proposed algorithm.

## Figures and Tables

**Figure 1 sensors-20-05363-f001:**
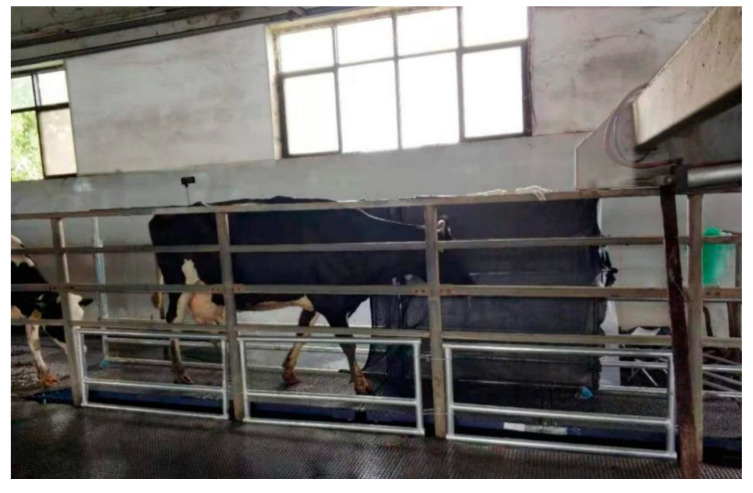
Cow weighing scene.

**Figure 2 sensors-20-05363-f002:**
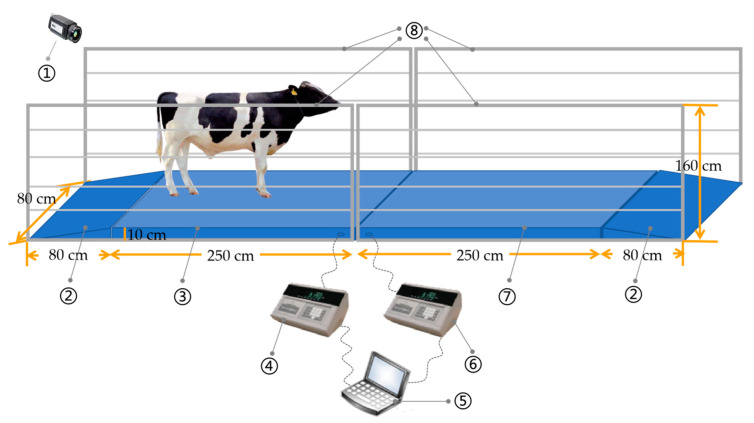
Cow weighing diagram: 1—camera; 2—slope; 3—dynamic weighing platform; 4—dynamic weighing indicator; 5—computer; 6—static weighing indicator; 7—static weighing platform; 8—guardrail.

**Figure 3 sensors-20-05363-f003:**
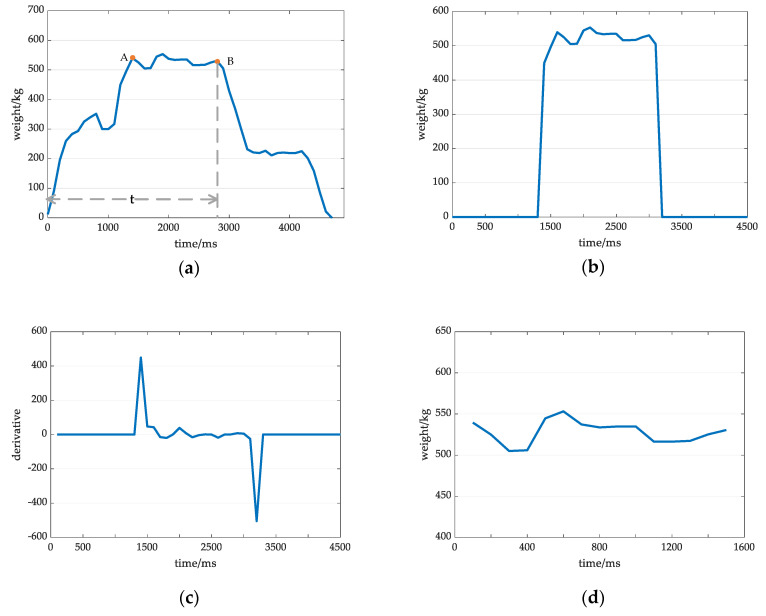
Effective signal acquisition process: (**a**) Original signal from cow weighing; (**b**) The signal processed based on the threshold K1; (**c**) The derivative of the signal in (**b**); and (**d**) Valid data.

**Figure 4 sensors-20-05363-f004:**
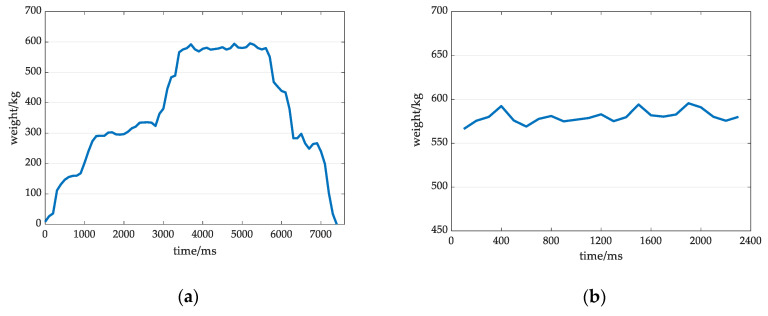
Original weight signal and valid signal under the slow grade of active degree conditions: (**a**) Original weight signal at a low grade; (**b**) Valid signal at a low grade.

**Figure 5 sensors-20-05363-f005:**
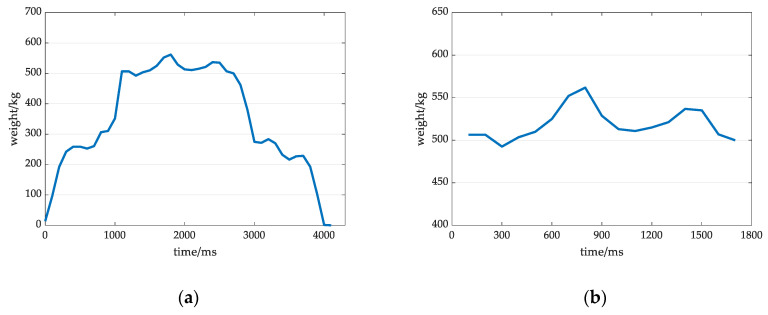
Original weight signal and valid signal at a medium grade of active degree: (**a**) Original weight signal at a medium grade; (**b**) Valid signal at a medium grade.

**Figure 6 sensors-20-05363-f006:**
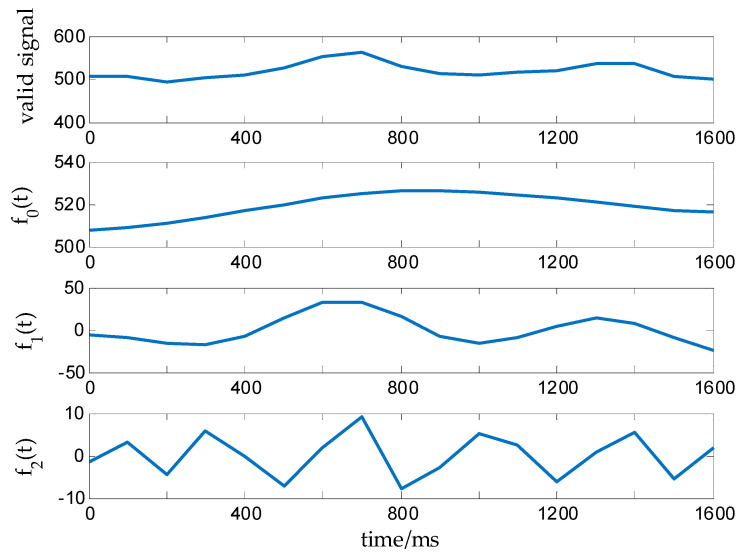
EWT decomposition diagram for a medium-grade activity signal.

**Figure 7 sensors-20-05363-f007:**
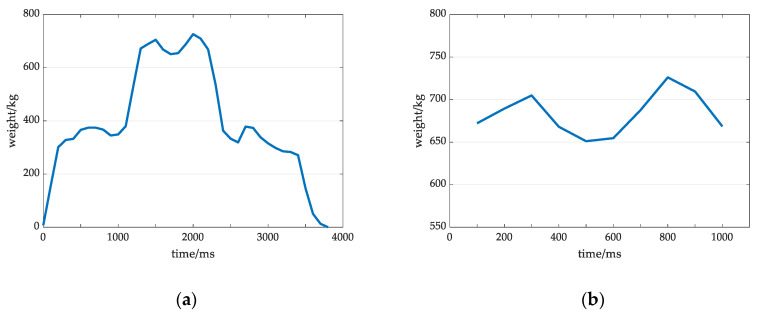
Original weight signal and valid signal in the high grade of active degree: (**a**) Original weight signal at a high grade; (**b**) Valid signal at a high grade.

**Figure 8 sensors-20-05363-f008:**
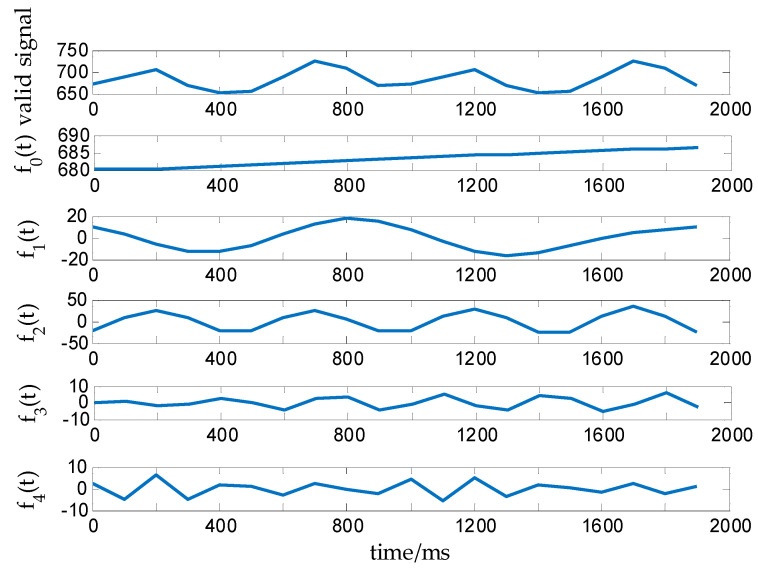
EWT decomposition diagram for a high-grade activity signal.

**Figure 9 sensors-20-05363-f009:**
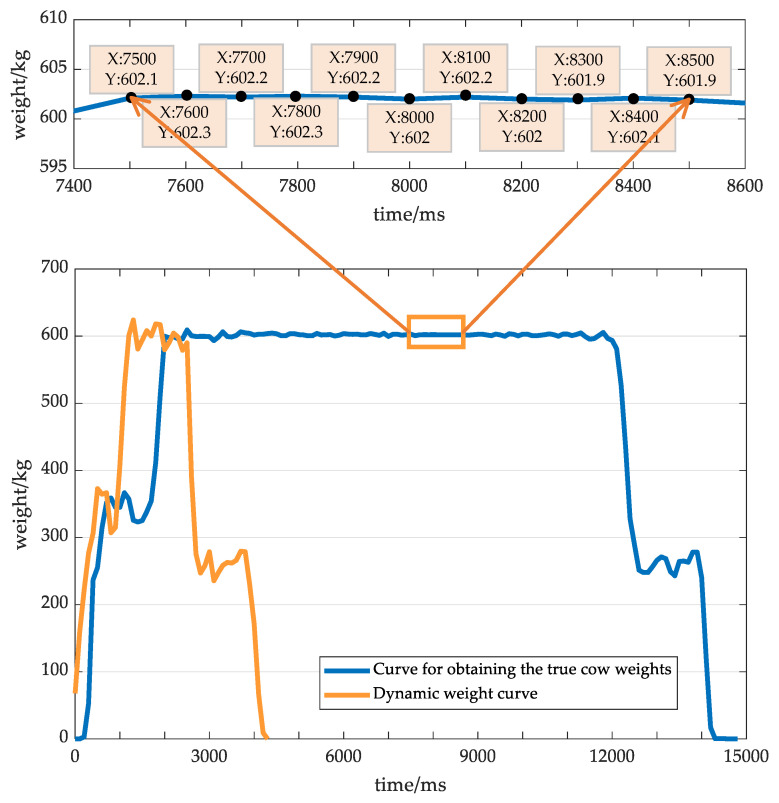
Curve for obtaining the true cow weights.

**Figure 10 sensors-20-05363-f010:**
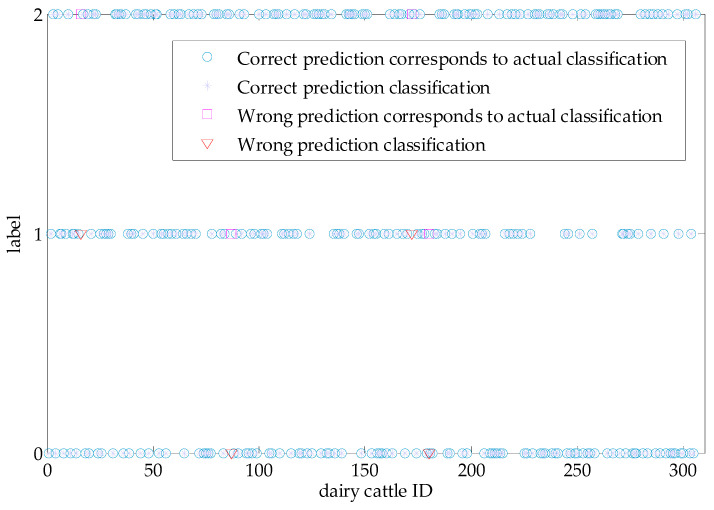
Actual and predicted classification diagram for the test set.

**Figure 11 sensors-20-05363-f011:**
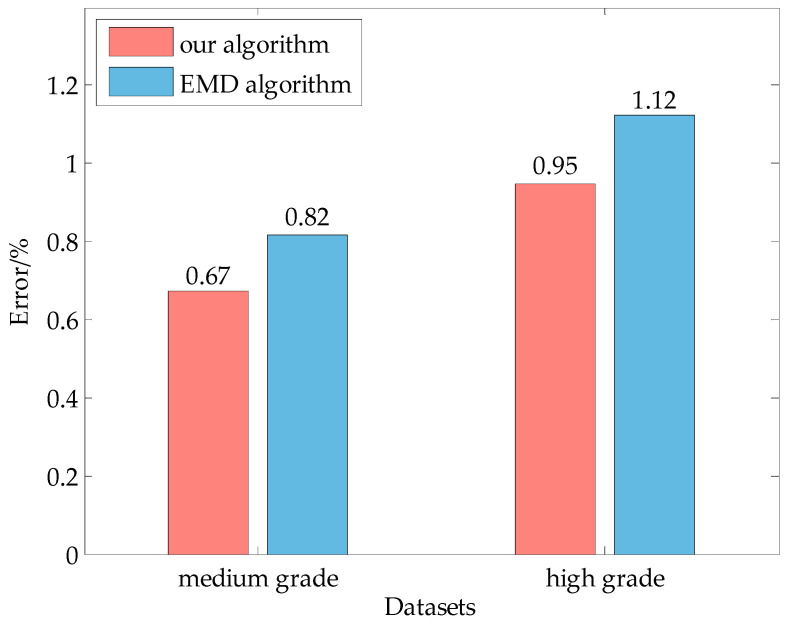
Comparison of the average error rate between the proposed algorithm and EMD algorithm.

**Table 1 sensors-20-05363-t001:** Average values for three motion states with different eigenvalues.

Characteristic	Category	Average
range/kg	low grade	24.3260
	medium grade	59.7984
	high grade	107.9911
standard deviation	low grade	10.5815
	medium grade	19.5883
	high grade	37.8103
peak factor	low grade	0.0650
	medium grade	0.1090
	high grade	0.1901
waveform factor	low grade	1.0002
	medium grade	1.0006
	high grade	1.0022
speed/m/s	slow grade	0.7525
	medium grade	1.2500
	high grade	1.6685

**Table 2 sensors-20-05363-t002:** Evaluation index formulas.

Evaluation Index	Computational Formula
Precision/%	$Precision=\frac{TP}{TP+FP}$
recall/%	$Recall=\frac{TP}{TP+FN}$
specificity/%	$Specificity=\frac{TN}{TN+FP}$
F1_score	$F1\text{\_}Score=\frac{2PPV*REC}{PPV+REC}$

**Table 3 sensors-20-05363-t003:** Three categories of the classification confusion matrix.

Predicted Classification	Actual Classification			
	Low Grade	Medium Grade	High Grade	Total
low grade	101	2	0	103
medium grade	0	88	2	90
high grade	0	0	113	113
total	101	90	115	306

**Table 4 sensors-20-05363-t004:** Evaluation index.

Evaluation Index	Category		
	Low Grade	Medium Grade	High Grade
accuracy/%	98.6928		
precision/%	98.0583	97.7778	100
recall/%	100	97.7778	98.2601
specificity/%	99.0244	99.0741	100
F1_score	0.9902	0.9778	0.9912

**Table 5 sensors-20-05363-t005:** Comparison of the average and maximum error rates for the three active degrees.

Motion State	Mean Error Rate/%	Maximum Error Rate/%
low grade	0.1838	0.5742
medium grade	0.6724	1.6924
high grade	0.9462	1.9696

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
