# Peer review of "Research on a Dynamic Algorithm for Cow Weighing Based on an SVM and Empirical Wavelet Transform"

_sensors, 2020, doi:10.3390/s20185363_

Round 1

Reviewer 1 Report

This paper proposes a dynamic algorithm for cow weighing based on SVM and empirical wavelet transform. This topic is practical and interesting. The reviewer’s comments are given below.

  1. The author should introduce that how to estimate the true weight of a cow by using the dynamic weighing data.
  2. How to distinguish between the slow and fast signal should be further clarified.
  3. It seems that the proposed algorithm does not need the data about the walking speed. Can the speed measuring device be replaced by an algorithm?
  4. The improvement of the proposed algorithm is not significant comparing with the existing EMD algorithm. Maybe speed is not the main factor causing measurement errors.

Reviewer 2 Report

The summary and conclusions should briefly outline the approach.

It seems strange in the graphs to present the units of measurement by a fraction.

The experimental equipment should be described in more detail: parameters, accuracy.

Round 2

Reviewer 1 Report

The author has addressed all the question of the reviewer,and the reviewer has no other questions.